# Modelling and correcting the impact of RF pulses for continuous monitoring of hyperpolarized NMR

Gevin von Witte[1,2], Matthias Ernst[2], and Sebastian Kozerke[1]

[1]University and ETH Zurich, Institute for Biomedical Engineering, Zurich 8092, Switzerland
[2]ETH Zurich, Laboratory of Physical Chemistry, Zurich 8093, Switzerland

**Correspondence:** Sebastian Kozerke (kozerke@biomed.ee.ethz.ch)

**Abstract.** Monitoring the build-up or decay of hyperpolarization in nuclear magnetic resonance requires radio-frequency (RF) pulses to generate observable nuclear magnetization. However, the pulses also lead to a depletion of the polarization and, thus, alter the spin dynamics. To simulate the effects of RF pulses on the polarization build-up and decay, we propose a first-order rate-equation model describing the dynamics of the hyperpolarization process through a single source and a relaxation term. The model offers a direct interpretation of the measured steady-state polarization and build-up time constant. Furthermore, the rate-equation model is used to study three different methods to correct the errors introduced by RF pulses: (i) a $1/\cos^{n-1}\theta$ correction ($\theta$ denoting the RF pulse flip angle), which is only applicable to decays, (ii) an analytical model introduced previously in the literature and (iii) an iterative correction approach proposed here. The three correction methods are compared using simulated data for a range of RF flip angles and RF repetition times. The correction methods are also tested on experimental data obtained with dynamic nuclear polarization (DNP) using 4-oxo-TEMPO in $^1$H glassy matrices. It is demonstrated that the analytical and iterative corrections allow to obtain accurate build-up times and steady-state polarizations (enhancements) for RF flip angles up to 25°during the polarization build-up process within $\pm 10\%$ error when compared to data acquired with small RF flip angles (<3°). For polarization decay experiments, corrections are shown to be accurate for RF flip angles up to 12°. In conclusion, the proposed iterative correction allows to compensate for the impact of RF pulses offering an accurate estimation of polarization levels, build-up and decay time constants in hyperpolarization experiments.

*Copyright statement.* TEXT

## 1 Introduction

Improving the sensitivity of nuclear magnetic resonance (NMR) through hyperpolarization methods (Ardenkjaer-Larsen et al., 2015; Kovtunov et al., 2018; Akbey et al., 2013; Corzilius, 2020) requires an understanding of the limiting processes and, hence, accurate experimental measurements and data. In dynamic nuclear polarization (DNP), repeated radio-frequency (RF) pulses are applied to measure build-up and decay times as well as steady-state polarization. However, the readout RF pulses alter the state of the spin system by converting some of the polarization into detectable transverse magnetization. The larger

the RF pulses, the stronger the polarization is affected by the measurement process and with this the time evolution of the system. This leads to changes in the experimentally determined parameters compared to the undisturbed situation where no

RF pulses are applied. The effect of RF pulses can be minimized by using small flip-angle pulses with long repetition times or by repeating DNP experiments with a single large flip-angle pulse applied at the end of the individual experiment. The former approach often leads to noisy data and, hence, to poor estimates of the build-up time constant and steady-state polarization, whereas the latter is time consuming. We investigate an alternative path with repeated pulses of intermediate RF flip angles and repetition times. We correct for the effect of the readout RF pulses on the spin dynamics, leading to more accurate and faster

measurements.

The manuscript is divided into two parts. First, different RF correction methods are investigated in simulations using a rate-equation model consisting of a single polarization source and a relaxation term. Second, the simulated RF correction approaches are tested experimentally on data obtained with DNP in glassy $^1$H matrices containing 4-oxo-TEMPO. Together, the theoretical foundation for the correction of RF pulse effects in hyperpolarized NMR and its practical feasibility are presented, showing

the benefit of larger RF flip angles to obtain more accurate measurements of the experimental quantities of interest.

## 2   Theory: Rate-equation model

Let us consider a system that includes a hyperpolarization source and a relaxation term. For the source, we start from Fermi's golden rule and assume that the injected polarization is proportional to the available density of states, with the rate constant given by $k_W$. The total density of states is denoted by $A$, the occupied states by the nuclear polarization $P$ and, hence, the

available density of states is given by $(A - P)$. The relaxation is characterized by the relaxation-rate constant $k_R$. In the following, we ignore the thermal-equilibrium polarization as it is typically small compared to the polarization generated by the hyperpolarization process, e.g., enhancements $\epsilon = \frac{P_{\text{hyp}}}{P_{\text{eq}}}$ of more than 100 are often reported (Ardenkjær-Larsen et al., 2003; Jähnig et al., 2017; Leavesley et al., 2018; Ni et al., 2013; Corzilius, 2020; Rej et al., 2015; Kwiatkowski et al., 2018a; Shimon et al., 2022; Yoon et al., 2019; Dementyev et al., 2008; Hope et al., 2021; Jardón-Álvarez et al., 2020). Combining the above

arguments, the rate equation for the polarization is given by

$$\frac{\mathrm{d}P}{\mathrm{d}t} = (A - P)k_W - k_R P. \tag{1}$$

In the following, $k_W$ will be referred to as the (DNP) polarization injection rate as we describe the model based on the experimental setup of DNP. However, it can also be adopted for spin-exchange optical pumping (SEOP) (Walker and Happer, 1997), para-hydrogen based techniques (Natterer and Bargon, 1997; Adams et al., 2009; Kovtunov et al., 2018), triplet DNP

in pentacene crystals as polarization sources for target solutions (Tateishi et al., 2014; Miyanishi et al., 2021; Eichhorn et al., 2022) or nitrogen-vacancy (NV) centers in diamond to hyperpolarize surface or bulk spins in diamond (King et al., 2015; Broadway et al., 2018; Ajoy et al., 2018; Miyanishi et al., 2021).

Here, $A$ describes the total density of states which are accessible for building up nuclear hyperpolarization $P$. In DNP, the magnitude of $A$ would be determined by the thermal electron polarization as this governs the maximally possible enhancement.

In spin-exchange optical pumping (SEOP), $A$ would be given by the polarization of alkali atoms under circular-polarized laser irradiation (Walker and Happer, 1997), in para-hydrogen-based techniques, such as para-hydrogen-induced hyperpolarization (PHIP) or signal amplification by reversible exchange (SABRE), by the initial polarization level of the para-hydrogen molecules (Natterer and Bargon, 1997; Adams et al., 2009; Kovtunov et al., 2018).

The mechanism of (DNP) polarization injection can be a complex problem as it not only involves the initial quantum-mechanical polarization transfer from the electron to a hyperfine-coupled nucleus but also strong paramagnetic relaxation and the transport of the created nuclear polarization from the nuclei close to the electron (local nuclei) into the bulk as discussed in Prisco et al. (2021). This spin transport might be drastically slowed down due to paramagnetic shifts of the local nuclei compared to the bulk. This aspect, often called spin-diffusion barrier, has recently received renewed interest (Smith et al., 2012; Wittmann et al., 2018; Wenckebach et al., 2021; Tan et al., 2019; Stern et al., 2021; Chessari et al., 2022). Our rate-equation model largely ignores these microscopic complications by describing the polarization injection as a single step that builds up the polarization. We will address the applicability of our proposed model to the various DNP mechanisms in the Discussion section.

Solving Eq. (1) leads to

$$P(t) = \frac{Ak_{\mathrm{W}}}{k_{\mathrm{W}} + k_{\mathrm{R}}} \left(1 - e^{-(k_{\mathrm{W}} + k_{\mathrm{R}})t}\right) \tag{2}$$

which can be compared to an experimentally used ansatz of the form

$$P_{\mathrm{exp}}(t) = P_0(1 - e^{-t/\tau_{\mathrm{bup}}}) \tag{3}$$

to find a correspondence between the parameters in our theoretical model and the phenomenological experimental description. For steady-state polarization $P_0$ and the build-up time constant $\tau_{\mathrm{bup}}$ one obtains

$$P_0 = \frac{Ak_{\mathrm{W}}}{k_{\mathrm{W}} + k_{\mathrm{R}}} = Ak_{\mathrm{W}}\tau_{\mathrm{bup}} \tag{4a}$$

$$\tau_{\mathrm{bup}}^{-1} = k_{\mathrm{W}} + k_{\mathrm{R}} \tag{4b}$$

and

$$k_{\mathrm{W}} = \tau_{\mathrm{bup}}^{-1}\frac{P_0}{A} \tag{5a}$$

$$k_{\mathrm{R}} = \tau_{\mathrm{bup}}^{-1}\left(1 - \frac{P_0}{A}\right). \tag{5b}$$

For an identical relaxation-rate constant $k_{\mathrm{R}}$, a smaller injection parameter $k_{\mathrm{W}}$ would lead to longer build-up times and lower enhancements. For a value of $k_{\mathrm{W}}$ much larger than $k_{\mathrm{R}}$ the steady-state polarization would approach $A$ and the build-up time would be a measure of the injection parameter. However, this scenario is rarely observed experimentally and would represent the ideal case. For rather small experimental polarizations with respect to $A$, the build-up time would be similar to the relaxation-rate constant. We note that similar expressions for the steady-state polarization and build-up time have been derived in (Smith et al., 2012; Corzilius et al., 2012) for coupled nuclear-electron rate-equation systems.

The model proposed above only requires three parameters to describe the build-up dynamics: $A$, $k_W$ and $k_R$. The value of $A$ is determined by experimental conditions, e.g. in DNP by the thermal electron polarization which depends mostly on the magnetic field and temperature. The rate constants $k_W$ and $k_R$ can be deduced from the measured build-up time constant and the steady-state polarization as indicated in Eqs. (5a, 5b).

    Eliminating the injection (source) term from Eq. 2 or setting $k_W$ to zero, leaves only the relaxation term. This corresponds to

a decay experiment which is described by a simple exponential decay $P_{\mathrm{exp,d}}(t) = P_0' e^{-t/\tau_{\mathrm{decay}}}$. The solution of the differential equation is straightforward and the decay time constant is given by

$$\tau_{\mathrm{decay}}^{-1} = k_R. \tag{6}$$

The initial polarization in the decay case is given by the polarization that was created during the hyperpolarization build-up. We would like to stress that the relaxation-rate constant during the decay does not have to be the same as during the build-up

since the experimental conditions may not be the same. For example, the microwave irradiation necessary for DNP is turned on during the build-up but is typically switched off during the decay measurements.

    In the following, the proposed rate-equation model is studied in simulations using a time slicing algorithm with RF pulses depleting the polarization repeatedly. Different methods to correct for the effects of RF pulses on the hyperpolarization dynamics are investigated theoretically before being tested experimentally.

## 3   Theory: Radio-frequency pulse correction

To investigate the effects of repeated RF pulses on the magnetization and the polarization dynamics, we integrate Eq. (1) and apply RF pulses (with flip angle $\theta$) at a fixed repetition time $T_R$. To avoid confusion, we do not specify a time unit in our simulations as different samples can have widely different time scales experimentally, e.g., $^1$H DNP with 4-oxo TEMPO builds up in tens of seconds (see experimental results below), $^{13}$C DNP in diamond through the endogenous P1 centers takes tens of

minutes (Kwiatkowski et al., 2018a) and silicon nano- and microparticles take hours (Dementyev et al., 2008; Kwiatkowski et al., 2018b). Fig. 1a shows simulated build-up curves under different RF readout schemes relative to a reference simulation without RF pulses. Stronger pulses or shorter repetition times lead to reduced apparent build-up times and steady-state polarization levels as shown in Tab. 1 and Fig. 1a.

| $\theta$ | [°] | 2.5 | 2.5 | 7 | 12.5 | 25 |
|---|---|---|---|---|---|---|
| $T_R$ | [a.u.] | 2 | 1 | 2 | 2 | 2 |
| $\tau_{\mathrm{bup}}$ | [a.u.] | 48.9 | 47.8 | 42.2 | 31.3 | 14.5 |
| $P_0$ | | 0.293 | 0.287 | 0.254 | 0.190 | 0.091 |

**Table 1.** Fitted build-up times of noiseless simulated data under the influence of different RF schemes (compare Fig. 1a). Assumed experimental parameters without pulses: $P_0 = 0.3$, $\tau_{\mathrm{bup}} = 50$, $A = 1$.

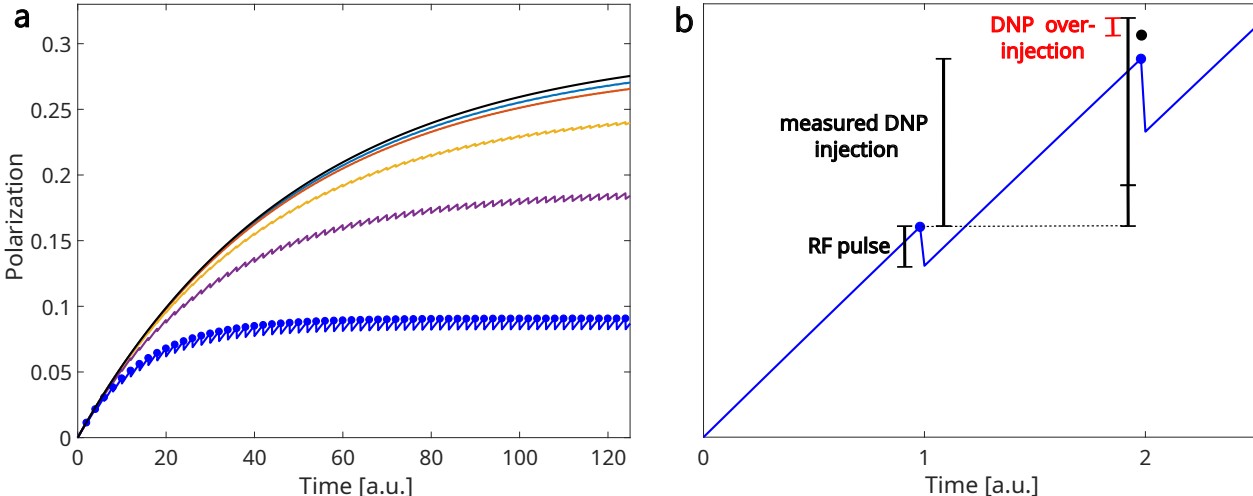

**Figure 1. (a)** Comparison of simulated build-up curves under the influence of RF pulses (see text for details). The black curve shows the build-up without RF pulses. The RF scheme for the other curves (from top to bottom): 2.5°, 2 time units; 2.5°, 1 time unit; 7°, 2 time units; 12.5°, 2 time units; 25°, 2 time units. Assumed experimental parameters without pulses: $P_0 = 0.3$, $\tau_{\mathrm{bup}} = 50$, $A = 1$. **(b)** Illustration of RF correction during build-up. The blue points indicate the measured polarization. The black point indicates the true polarization in the absence of RF pulses. Note that the first data point is exact without any RF correction. The blue line shows the polarization in the presence of RF pulses, DNP injection and relaxation. An increased signal due to DNP injection is observed from the first to the second data point. The RF pulse decreases polarization.

Correcting the effects of RF pulsing during a build-up requires to consider three aspects as outlined in Fig. 1b: (i) the measured polarization might change between consecutive data points as the steady-state is not yet reached, (ii) a readout RF pulse reduces the polarization while the polarization is assumed to be unaffected ($M_{xy} = \sin(\theta)M_z$) and (iii) the reduced polarization leads to a weaker effect of relaxation and a stronger effect of polarization injection.

In the following, an iterative correction algorithm is proposed that takes the measured data as input and corrects for the effects of repeated RF pulses. The first two terms of the correction algorithm describe the measured polarization difference between consecutive data points and the correction for the depletion by an RF pulse. The third contribution, which we call $\Delta_{n-1}$ for the $n$-th acquired data point, describes the DNP overinjection due to the changes in polarization given the $(n-1)^{\mathrm{th}}$ RF pulse. In the following, we will denote the measured polarization without any correction by $P_n$ and the corrected polarization by $\tilde{P}_n$, equal to the theoretical RF-free experiment. The DNP overinjection $\Delta_{n-1}$ is given by

$$d\tilde{P}_{n-1} = \left[(A - \tilde{P}_{n-1})k_{\mathrm{W}} - k_{\mathrm{R}}\tilde{P}_{n-1}\right]dt$$

$$dP_{n-1} = \left[(A - \cos(\theta)P_{n-1})k_{\mathrm{W}} - k_{\mathrm{R}}\cos(\theta)P_{n-1}\right]dt$$

$$\Rightarrow \Delta_{n-1} = (\tilde{P}_{n-1} - \cos(\theta)P_{n-1})\underbrace{(k_{\mathrm{W}} + k_{\mathrm{R}})}_{\tau_{\mathrm{bup}}^{-1}}dt \tag{7}$$

and with this we can write an iterative correction

$$\tilde{P}_n = \tilde{P}_{n-1} + \underbrace{(P_n - P_{n-1})}_{\text{DNP injection}} + \underbrace{(\cos(\theta)^{-1} - 1)\cos(\theta)P_{n-1}}_{\text{RF pulse}}$$

$$- \underbrace{(\tilde{P}_{n-1} - \cos(\theta)P_{n-1})(k_{\mathrm{W}} + k_{\mathrm{R}})T_{\mathrm{R}}}_{\text{DNP overinjection through RF depleted polarization } (\Delta_{n-1})}$$

$$= \tilde{P}_{n-1} + (P_n - \cos(\theta)P_{n-1})$$

$$- (\tilde{P}_{n-1} - \cos(\theta)P_{n-1})(k_{\mathrm{W}} + k_{\mathrm{R}})T_{\mathrm{R}}. \tag{8}$$

We use the definition of the build-up time constant from Eq. (4b) as already indicated in Eq. (7). To include an RF-corrected value of the build-up time, we use the analytic model presented in the supplementary information of (Capozzi et al., 2017). Accordingly, a build-up time constant $\tau_{\mathrm{bup}}$ (or alternatively the decay time constant $\tau_{\mathrm{decay}}$) can be calculated according to

$$\tau_{\mathrm{bup}} = \left(\frac{1}{\tau'} + \frac{\ln(\cos(\theta))}{T_{\mathrm{R}}}\right)^{-1} \tag{9}$$

with $\tau'$ being the measured time constant without any correction for RF pulses. This approach considers RF pulsing as an external apparent relaxation channel. As the model was introduced by Capozzi and Comment et al., we will refer to it as the CC-model in the following.

Based on our rate-equation model and the notion that the relative change of the steady-state polarization with RF pulsing is
only due to the change in build-up time (compare Eq. (4a) and Tab. 1), the CC-model can be extended to provide also corrected values for the steady-state polarization according to:

$$P_0 = P_0' \frac{\tau_{\mathrm{bup}}}{\tau'} \tag{10}$$

with $\tau'$ and $P_0'$ being the measured, uncorrected build-up time and steady-state polarization and $\tau_{\mathrm{bup}}$ the CC-corrected build-up time constant. Conceptually, this can be understood as the injection rate constant $k_{\mathrm{W}}$ being undisturbed by the RF pulses
while the observed relaxation-rate constant $k_{\mathrm{R}}$ appears increased by the RF pulses. We note that relaxation in NMR usually describes incoherent processes, while RF pulses induce a coherent process. Assuming large hyperpolarization enhancements, such that the thermal polarization can be neglected, incoherent spin lattice relaxation drives the polarization back to zero or more precisely to the (negligible) thermal equilibrium. Hence, RF pulses and incoherent relaxation processes have the same effect on the hyperpolarization. In the following, we will use the term "apparent relaxation due to RF perturbations"

to refer to the polarization-depleting rate of RF pulses, indicating that they have a similar effect to spin-lattice relaxation in hyperpolarization but not being an incoherent relaxation process.

A third method to correct for the readout RF pulses is given by

$$1/\cos^{n-1}\theta \tag{11}$$

with $n$ being the number of RF pulses. However, this method is only applicable to decays.

## 4 Methods

Simulations and computational corrections were implemented in Matlab (Mathworks, Natick, MA). All experimental data were acquired with a 50mM 4-oxo-TEMPO in water/glycerol mixtures using DNP. In particular, we compare two different sample formulations with TEMPO in DNP juice (6:3:1 mixture of glycerol-$d_8$, $D_2O$ and $H_2O$) or TEMPO in $(1/1)_V$ $H_2O$/ glycerol (no deuteration, all natural abundance). After mixing the ingredients, the filled sample container was frozen in liquid nitrogen before being transferred to a cryogenically pre-cooled polarizer (cryostat temperature during the transfer below 20K).

The natural abundance sample formulation was reported to show a mono-exponential build-up in our 7 T setup (299 MHz $^1$H Larmor frequency) (Jähnig et al., 2019). In addition, fast proton spin diffusion and a homogeneous radical distribution should ensure a homogeneous mono-exponential build-up and decay of the polarization. Compared to our previously published work, we upgraded the system to a new microwave source (200 mW, Virginia Diodes Inc., USA) and silver-plated the wave-guides to reduce resistive losses, yielding approximately eight times more microwave power as before in the sample space (around 65 mW) (Himmler et al., 2022). Details of the experimental setup can be found elsewhere (Jähnig et al., 2017; Himmler et al., 2022). The NMR measurements were performed at a sample temperature of 3.3 K with a Bruker Avance III HD (Bruker BioSpin AG, Switzerland) spectrometer. A prescan delay of 18 µs was used to protect the spectrometer from signal overflow. All data processing was performed using Matlab scripts.

## 5 Results

### 5.1 Simulations

A comparison of simulated noise-free and noisy data without correction and with iterative correction is shown in Fig. 2. For larger flip angles, the uncorrected build-up deviates from the theoretical value without RF pulsing. Employing the iterative correction for noise-free data works accurately up to the largest flip angles tested (37°). Introducing noise into the simulations, leads to a small deviation for the largest flip angle considered.

To study the performance of the corrections more systematically, we performed the corrections 5000 times for each $\theta$-$T_R$ pair considered. The results of these simulations are shown in the Supplementary Information (section S1 for build-up curves, section S2 for decay curves), yielding similar accuracy and precision for the CC-model and the iterative correction.

The $1/\cos^{n-1}$ correction for the decay curves performs similar to the other two methods (see Discussion).

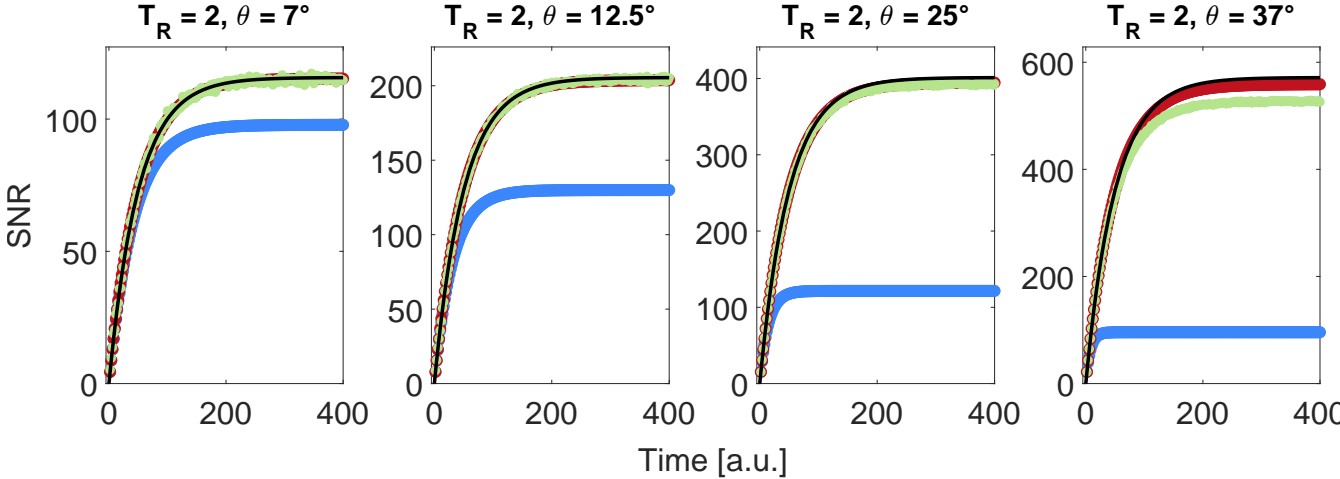

**Figure 2.** Comparison of noise-free uncorrected (blue), iteratively corrected noise-free (red) and noisy (green) data together with the exponential build-up function (black) given in Eq. 3. Assumed experimental parameters without pulses: $P_0 = 0.3$, $\tau_{\mathrm{bup}} = 50$, $A = 1$, noise = $3.2 \cdot 10^{-4}$.

In addition to studying the accuracy and precision depending on the flip angle and repetition time, we simulated the SNR-dependence of the iterative and CC-model. For this, we varied the steady-state polarization as well as the noise in 10 steps each and used all combinations of the two parameters. The results for these simulations are shown in Fig. 3. For both corrections a minimum SNR of around 5 is found with slightly higher values for large flip angles (25°) to avoid a deviation of the parameters by more than 10% from the values without RF pulsing. SNR in this context refers to the SNR at the steady-state polarization of the uncorrected build-up.

## 5.2 Experiment

Experimental build-up curves acquired with different flip angles are shown in Fig. 4a together with an example of the iterative correction and a simulation of the rate-equation model confirming the validity of our approach. The input parameters of the simulated build-up curves are derived from the experimentally measured steady-state polarization and build-up time constant using Eqs. (5a) and (5b). The parameter $A$ was set to the thermal electron polarization of 89%. The estimated relaxation-rate constant of $0.024\mathrm{s}^{-1}$ for the build-up was much larger than the measured decay-rate constant of $0.006\mathrm{s}^{-1}$. A typical decay measurement before and after correction for RF pulses is shown in Fig. 4b.

We first performed small flip angle measurements (for both samples separately) since the measured parameters under these conditions are very close to the unperturbed case (cf. Tab. 1). After these calibration measurements, we performed measurements with larger flip angles and different repetition times to estimate the range over which the corrections perform accurately in experimental data. The results for all measurements with TEMPO in DNP juice are summarized in Tab. 2 and Fig. 5. The

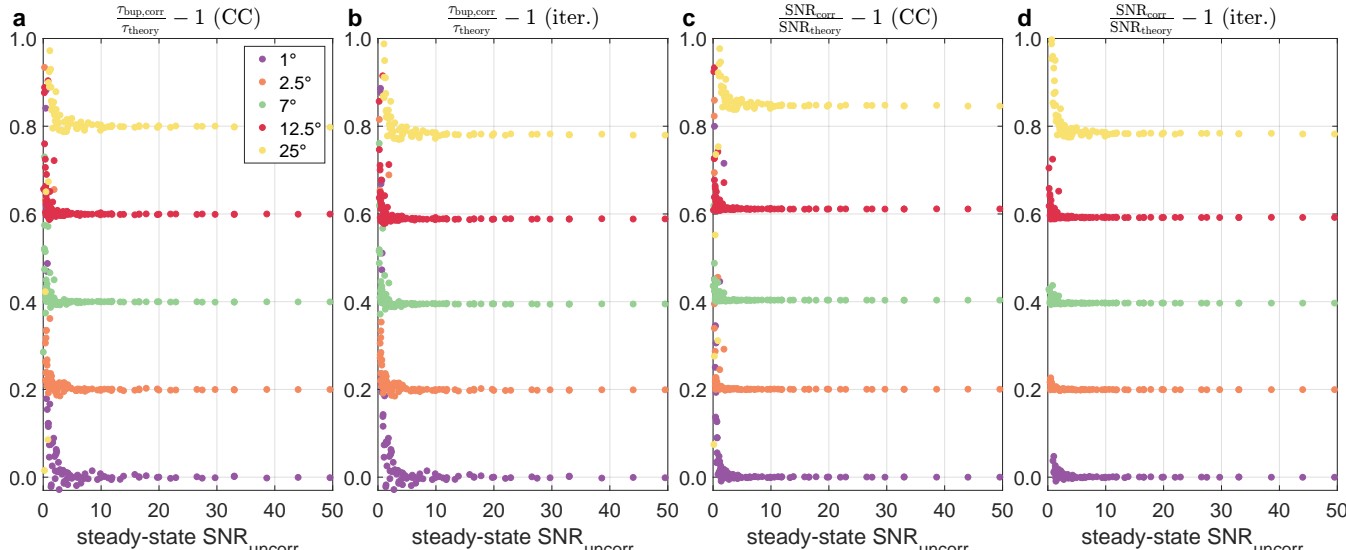

**Figure 3.** Minimum SNR required for the CC-model ((**a**) for $\tau_{\mathrm{bup}}$ and (**c**) SNR (polarization)) and iterative correction ((**b**) for $\tau_{\mathrm{bup}}$ and (**d**) SNR (polarization)) for different flip angles, all with $T_{\mathrm{R}} = 2$. The accuracy w.r.t. theoretical parameters is shown against the measured, uncorrected steady-state SNR. Different flip angles are vertically offset (0.2) for clarity. Simulation parameters: $\tau_{\mathrm{bup}} = 50$, $A = 1$, $P_0$ was varied in ten steps between 0.01 and 0.1 while the noise was varied in ten steps between $3.2 \cdot 10^{-5}$ and $3.2 \cdot 10^{-3}$. 100 noisy build-ups were simulated and the corrected parameters averaged for each data point displayed. Errorbars are omitted for clarity as these would hide all low SNR data.

respective data sets of the natural abundance sample are shown in Tab. 3 and Fig. 6, described by an apparent relaxation-rate due to RF perturbations, given by $\sin(\theta)/T_{\mathrm{R}}$.

For larger flip angles and more repeat pulses, the measured uncorrected parameters deviate from the values obtained in the calibration measurements. However, the corrected parameters give accurate values compared to the calibration measurements ($\pm 10\%$). In particular, build-up curves can be corrected with the CC-model and iterative correction up to $25°$ RF flip angles in our experiments. For the decay, the corrections become inaccurate earlier which will be discussed below: the $1/\cos^{n-1}$ correction becomes inaccurate for $5°$ pulses in our case (see Tab. 2), although its accuracy might be similar to the other two corrections with flip angles up to $12°$ possible (see Tab. 3). We note that the corrections can give accurate results with measured decay time constants of less than one-fifth of the expected value.

## 6 Discussion

We have demonstrated that the proposed rate-equation model allows to obtain corrected build-up times and steady-state polarization levels even for large RF flip angles ($25°$) during $^1$H (TEMPO in water/glycerol) polarization build-up yielding results with $\pm 10\%$ error compared to data acquired with small RF flip angles ($< 3°$). Based on simulations with added noise (see

| | $\theta$ [°] | $T_R$ [s] | $\sin(\theta)/T_R$ [$10^{-2}$ s$^{-1}$] | $\epsilon$ uncorr. | $\epsilon$ iter. | $\epsilon$ CC | $\tau_{\text{bup}}$ [s] uncorr. | $\tau_{\text{bup}}$ [s] iter. | $\tau_{\text{bup}}$ [s] CC | $\tau_{\text{decay}}$ [s] uncorr. | $\tau_{\text{decay}}$ [s] iter. | $\tau_{\text{decay}}$ [s] CC | $\tau_{\text{decay}}$ [s] $1/\cos^{n-1}$ |
|---|---|---|---|---|---|---|---|---|---|---|---|---|---|
| 1 | 0.7 | 5 | 0.3 | 142 | 142 | 142 | 31 | 31 | 31 | 173 | 173 | 173 | 173 |
| 2 | 0.7 | 2 | 0.6 | 142 | 142 | 142 | 31 | 31 | 31 | 172 | 174 | 174 | 173 |
| 3 | 0.7 | 1 | 1.3 | 141 | 141 | 141 | 30 | 30 | 30 | 167 | 170 | 170 | 170 |
| 4 | 0.7 | 0.5 | 2.5 | 140 | 141 | 141 | 30 | 30 | 30 | 166 | 170 | 170 | 170 |
| 5 | 1.5 | 2 | 1.3 | 134 | 135 | 135 | 30 | 30 | 30 | 167 | 172 | 172 | 171 |
| 6 | 1.5 | 1 | 2.7 | 133 | 135 | 135 | 30 | 30 | 30 | 161 | 171 | 171 | 170 |
| 7 | 1.5 | 0.5 | 5.3 | 132 | 134 | 134 | 29 | 30 | 30 | 152 | 171 | 171 | 169 |
| 8 | 2.4 | 5 | 0.9 | 133 | 134 | 134 | 31 | 31 | 31 | 171 | 176 | 176 | 176 |
| 9 | 2.4 | 2 | 2.1 | 131 | 133 | 133 | 30 | 30 | 30 | 162 | 175 | 175 | 173 |
| 10 | 2.4 | 1 | 4.3 | 129 | 133 | 133 | 29 | 30 | 30 | 149 | 172 | 172 | 169 |
| 11 | 2.4 | 0.5 | 8.5 | 126 | 133 | 133 | 29 | 30 | 30 | 131 | 173 | 173 | 167 |
| 12 | 4.7 | 2 | 4.1 | 127 | 133 | 133 | 29 | 30 | 30 | 135 | 175 | 174 | 169 |
| 13 | 4.7 | 1 | 8.2 | 121 | 132 | 133 | 27 | 30 | 30 | 110 | 175 | 175 | 163 |
| 14 | 4.7 | 0.5 | 16 | 110 | 132 | 133 | 25 | 30 | 30 | 801 | 76 | 176 | 150 |
| 15 | 7.1 | 2 | 6.2 | 117 | 130 | 130 | 27 | 30 | 30 | 105 | 177 | 176 | 162 |
| 16 | 7.1 | 1 | 12 | 106 | 129 | 130 | 24 | 29 | 29 | 75 | 178 | 178 | 147 |
| 17 | 7.1 | 0.5 | 25 | 89 | 128 | 129 | 20 | 29 | 29 | 48 | 183 | 184 | 109 |
| 18 | 12.2 | 2 | 11 | 97 | 127 | 130 | 22 | 29 | 29 | 60 | 189 | 189 | 130 |
| 19 | 12.2 | 1 | 21 | 78 | 127 | 130 | 18 | 29 | 29 | 36 | 193 | 190 | 73 |
| 20 | 12.2 | 0.5 | 42 | 56 | 129 | 131 | 13 | 29 | 30 | 20 | 246 | 267 | 7 |
| 21 | 24.7 | 2 | 21 | 54 | 124 | 133 | 12 | 30 | 30 | 20 | 304 | 469 | $7 \cdot 10^5$ |
| 22 | 24.7 | 1 | 42 | 34 | 137 | 145 | 8 | 33 | 33 | 11 | $1 \cdot 10^6$ | -375 | $5 \cdot 10^5$ |

**Table 2.** Overview of different experimental flip angles and correction methods with TEMPO in DNP juice. The iterative and CC-model are applicable to build-up and decay. refers to the CC-model. For the decay, we compare these two with a simple $1/\cos^{n-1}$ correction. $\epsilon$ refers to the DNP enhancement. $\sin(\theta)/T_R$ can be interpreted as an apparent relaxation-rate due to RF perturbations. This data is summarized in Fig. 5.

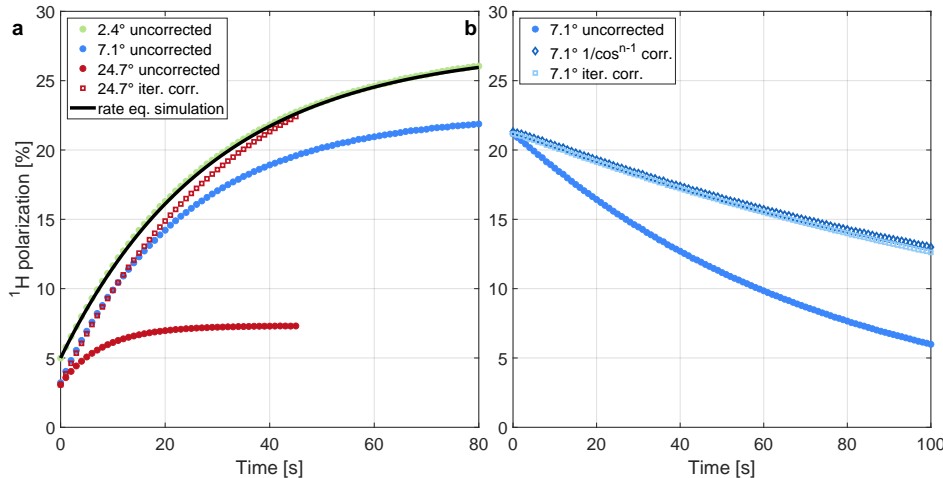

**Figure 4. (a)** Experimental build-ups with different flip angles (filled dots, see experiments 10, 16 and 22 in Tab. 2 for more details). For the 2.4° measurement, the corresponding build-up simulation is based on the thermal electron polarization $A = 0.89$, measured steady-state polarization and build-up time constant (see Eqs. (1,5a, 5b)). For the first data point of the simulation, the starting polarization is set to the first experimental data point as this initial polarization is an artefact of the measurement process (see Discussion for details). **(b)** Uncorrected and corrected decay under pulses with a flip angle of 7° every 1 s (exp. 16 in Tab. 2).

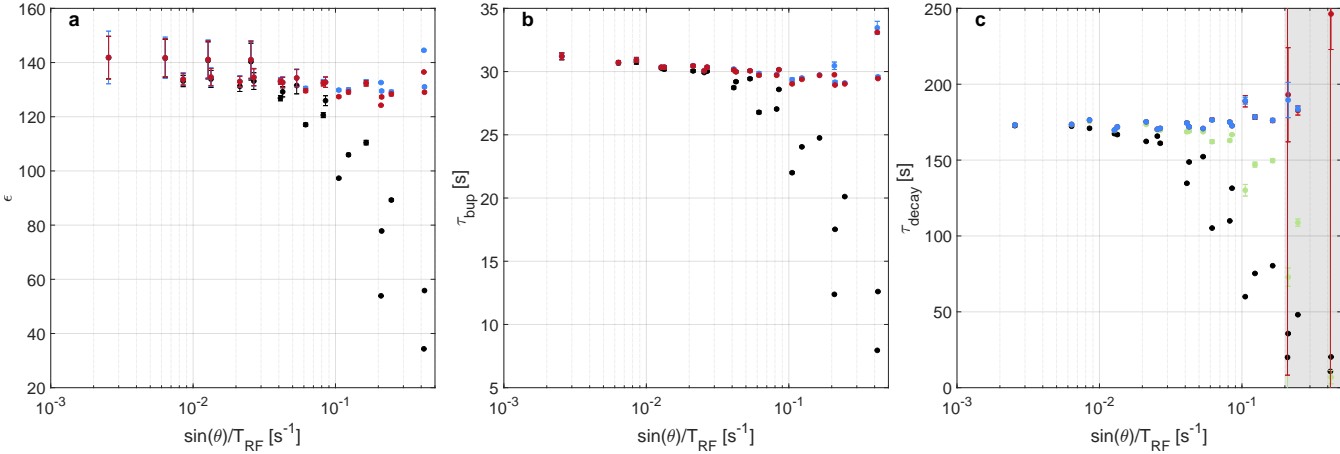

**Figure 5.** Experimental parameters - enhancement **(a)**, build-up **(b)** and decay times **(c)** - with and without correction for the different experiments with TEMPO in DNP juice as shown in Tab. 2, ordered by the apparent relaxation-rate due to RF perturbations $(\sin(\theta)/T_R)$. Black refers to the uncorrected data, red, blue and green to the iterative, CC and $1/\cos^{n-1}$ correction, respectively. The uncertainties extracted from the 95% fit intervals of the respective build-up and decay measurements are often smaller than the symbol.

205    Supplementary Information, sections S1 and S2 for build-up and decay curves, respectively), we expect the corrections to become inaccurate for too large flip angles (and apparent relaxation-rate due to RF perturbations). Experimentally, the cor-

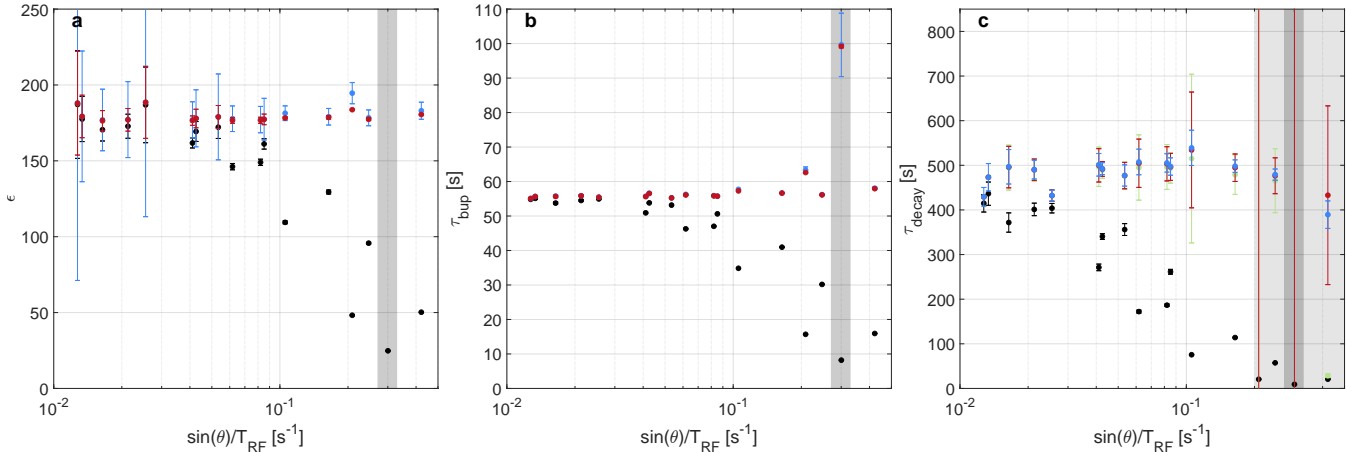

**Figure 6.** Experimental parameters - enhancement **(a)**, build-up **(b)** and decay times **(c)** - with and without correction for the different experiments with TEMPO in the natural abundance sample as shown in Tab. 3, ordered by the apparent relaxation-rate due to RF perturbations $(\sin(\theta)/T_R)$. Black refers to the uncorrected data, red, blue and green to the iterative, CC and $1/\cos^{n-1}$ correction, respectively. The uncertainties extracted from the 95% fit intervals of the respective build-up and decay measurements are often smaller than the symbol.

rections become inaccurate for build-ups acquired with flip angles between 25 and 37°, both for the CC-model and iterative correction. For decays, corrections fail earlier: between 5 and 12° for the $1/\cos^{n-1}$ model as well as between 12 and 25° for the CC-model and iterative correction. The lower accuracy of the decay can be attributed to a combination of reasons: (i) once

210 the apparent relaxation due to RF perturbations becomes much faster than the thermal relaxation, only few data points can be acquired to estimate the thermal rate constant as the hyperpolarized polarization is decaying fast; (ii) during the build-up, strong apparent relaxation due to RF perturbations is not the only contribution to the system dynamics as the (DNP) injection term gets also larger due to the lower polarization under RF pulsing. When the two reach a balance, the internal system dynamics is still important. In the decay case, the only large term dominating every other process is the apparent relaxation due

215 to RF perturbations, rendering the thermal relaxation a small perturbation; (iii) the decay measurement starts with a low initial polarization as the strong apparent relaxation due to RF perturbations caused the polarization at the end of the build-up to be small since we performed build-up and decay measurements in one experiment. This limits the number of data points with sufficient SNR for the overall decay fit to only a few points as the polarization is very quickly depleted due to RF pulses.

In general, the performance of the CC-model and iterative correction is identical with the latter having the additional ability

220 of correcting individual data points.

The $1/\cos^{n-1}$ correction works well for high SNR decay measurements but cannot be used for the build-ups due to the divergent nature of the correction factor. Furthermore, for low SNR, the results are inaccurate as the correction factor acts only on a single data point and amplifies the noise. The failure of the $1/\cos^{n-1}$ correction for the DNP juice sample already at 5° (compare Tab. 2) is partially related to the measurement process but mostly inherent to the single data point dependence of

225 this correction. For DNP juice, the spin-lattice $T_1$ relaxation time is much shorter than for the natural abundance sample. For

| | $\theta$ [°] | $T_R$ [s] | $\sin(\theta)/T_R$ [$10^{-2}$ s$^{-1}$] | $\epsilon$ uncorr. | $\epsilon$ iter. | $\epsilon$ CC | $\tau_{\text{bup}}$ [s] uncorr. | $\tau_{\text{bup}}$ [s] iter. | $\tau_{\text{bup}}$ [s] CC | $\tau_{\text{decay}}$ [s] uncorr. | $\tau_{\text{decay}}$ [s] iter. | $\tau_{\text{decay}}$ [s] CC | $\tau_{\text{decay}}$ [s] $1/\cos^{n-1}$ |
|---|---|---|---|---|---|---|---|---|---|---|---|---|---|
| 1 | 0.7 | 1 | 1.3 | 187 | 188 | 188 | 55 | 55 | 55 | 415 | 429 | 429 | 429 |
| 2 | 0.7 | 0.5 | 2.5 | 187 | 189 | 189 | 55 | 55 | 55 | 404 | 432 | 432 | 432 |
| 3 | 1.5 | 2 | 1.3 | 178 | 179 | 179 | 55 | 56 | 56 | 436 | 473 | 473 | 473 |
| 4 | 1.5 | 0.5 | 5.3 | 172 | 179 | 179 | 53 | 55 | 55 | 356 | 477 | 477 | 476 |
| 5 | 2.4 | 2 | 2.1 | 173 | 177 | 177 | 54 | 56 | 56 | 401 | 490 | 490 | 489 |
| 6 | 2.4 | 1 | 4.3 | 169 | 178 | 178 | 54 | 57 | 57 | 340 | 492 | 492 | 489 |
| 7 | 2.4 | 0.5 | 8.5 | 161 | 177 | 177 | 51 | 56 | 56 | 261 | 496 | 497 | 494 |
| 8 | 4.7 | 5 | 1.6 | 171 | 176 | 177 | 54 | 56 | 56 | 372 | 496 | 497 | 495 |
| 9 | 4.7 | 2 | 4.1 | 162 | 177 | 177 | 51 | 56 | 56 | 271 | 500 | 501 | 497 |
| 10 | 4.7 | 1 | 8.2 | 149 | 177 | 177 | 47 | 56 | 56 | 187 | 503 | 505 | 496 |
| 11 | 4.7 | 0.5 | 16 | 129 | 179 | 179 | 41 | 57 | 57 | 114 | 495 | 498 | 479 |
| 12 | 7.1 | 2 | 6.2 | 146 | 177 | 178 | 46 | 56 | 56 | 172 | 505 | 507 | 495 |
| 13 | 7.1 | 0.5 | 245 | 96 | 177 | 178 | 30 | 56 | 56 | 57 | 477 | 479 | 465 |
| 14 | 12.2 | 2 | 11 | 109 | 178 | 181 | 35 | 57 | 58 | 76 | 534 | 539 | 515 |
| 15 | 12.2 | 0.5 | 42 | 50 | 181 | 183 | 16 | 58 | 58 | 21 | 433 | 389 | 29 |
| 16 | 24.7 | 2 | 21 | 48 | 184 | 195 | 16 | 63 | 63 | 21 | 1595 | 1240 | $1 \cdot 10^6$ |
| 17 | 36.9 | 2 | 30 | 25 | 269 | 302 | 8 | 99 | 100 | 9 | $8 \cdot 10^5$ | -367 | $8 \cdot 10^4$ |

**Table 3.** Overview of different experimental flip angles and correction methods with TEMPO in a natural abundance $H_2O$/ glycerol sample. The iterative and CC-model are applicable to build-up and decay. For the decay, we compare these two with a simple $1/\cos^{n-1}$ correction. $\epsilon$ refers to the DNP enhancement. $\sin(\theta)/T_R$ can be interpreted as an apparent relaxation-rate due to RF perturbations. This data is summarized in Fig. 6.

both samples, data was acquired until there was either only a thermal signal remaining or if several hundred seconds elapsed. If the signal approaches the thermal signal generated between subsequent acquisitions, the $1/\cos^{n-1}$ correction would give an increasing signal as the correction factor diverges while the signal remains constant. With a careful selection of the number of data points acquired or analyzed, this problem could be mitigated. The complete failure of the $1/\cos^{n-1}$ correction (Fig. S8) for flip angles of 25° and more can be explained as follows. If the decay curve is sampled much longer than the decay time under RF pulsing, many data points with noise only are acquired. This noise is subsequently amplified by the divergent correction factor, leading to signals much larger than at the beginning of the decay, spoiling the exponential fit of the data. Again, this could be mitigated with a careful selection of the number of data points. These problems are not encountered for the other corrections as these rely on a fit of the complete data set and do not require manual user selection of data points to be included into the analysis, representing a major advantage for the automatic analysis of larger data sets.

The noise added in our simulations is relatively large compared to the noise measured in our experiments. In the simulations, a 2.5° pulse yielded an SNR of around 40 (see Fig. 2) while experimentally the SNR based on the first point of the FID was above 1000 (see Figs. S9 and S10) for a 2.4° flip angle for both samples. The lower SNR for the natural abundance sample compared to DNP juice is a result of the long prescan delay (18 μs) and short $T_2^*$ (see Fig. S9) resulting from the stronger proton-proton dipolar interactions. The simulations shown in Fig. 3 indicate that the corrections work even for low SNR measurements, i.e. an SNR of around 5. In these simulations, a slightly larger minimal SNR for the 25° pulses is observed. This might be due to the lower number of meaningful data points acquired as a result of the short apparent build-up time (compare Tab. 1). This might mean that in experiments with few pulses w.r.t. the build-up time constant, a higher minimal SNR is needed. Additionally, the simulations shown in Fig. 3 represent an average over 100 simulations for each displayed data point as these should demonstrate the accuracy of the corrections and not of the underlying build-up curve. Acquiring build-up curves with a steady-state SNR close to the theoretical minimum results in large uncertainties of the apparent build-up parameters, translating into inaccurate values for a single corrected build-up although the average over a number of build-up measurements would be corrected accurately.

In our analysis of the experimental data, we included an offset for the build-up and decay fit as a free fitting parameter. This was necessary as the first data point was acquired with some delay due to the time the spectrometer needs to load the data acquisition sequence after the separate saturation sequence (the start of a new experiment takes a few seconds). This leads to a higher polarization of the first acquired data point as can be seen in Fig. 4a (for the shown build-up simulation, the initial polarization of the first data point was set to the first experimental data point). Including this offset leads to very accurate (build-up) fits and with it of the RF correction. Including the offsets increases stability of the fits and corrections at the expense of larger uncertainties in the fitted parameters given the additional unknown.

The measured enhancements depend on a thermal equilibrium measurement. Since the presented results compare the relative differences between measurements, the uncertainty of the thermal equilibrium measurement does not affect the performance of the corrections. Furthermore, the conversion of the measured signal into enhancements depends on the flip angle. Uncertainty in the flip angles was not included in the calculation of the error bars. Another experimental complication causing differences between the experiments are drifts in the microwave power and cryostat as well as sample temperature. However, these are difficult to quantify but can be observed experimentally.

While our iterative correction approach permits data acquisition of hyperpolarized samples with relatively high SNR given that larger RF flip angles can be used, it is noted that it remains limited to samples which can be described by a single compartment. Violation of this assumption would lead to erroneous parameter estimation. Spin noise spectroscopy (McCoy and Ernst, 1989; Pöschko et al., 2015, 2016) is not limited by such constraints and might represent an alternative to pulsed measurements although SNR and duration of the experiment need to be considered.

Before concluding, we would like to discuss the validity of the proposed single-compartment rate-equation model for DNP in more detail. For solid effect (SE) DNP, the DNP injection into the bulk can be understood in terms of the polarization transfer from an electron to a hyperfine-coupled nucleus. From this polarized nucleus, the polarization spreads into the bulk through spin diffusion. The injection rate $k_{\mathrm{W}}$ can be seen as the overall rate for this joint process yielding a detectable magnetization

created in the bulk of the sample. Switching off the microwave would interrupt the hyperfine-mediated polarization transfer, causing a vanishing $k_{\mathrm{W}}$.

Contrary to the quantum mechanical description of the solid effect, thermal mixing (TM) DNP is modelled using a bath model with different temperatures for different spin systems. Such a spin bath model was used in previous work to characterize DNP processes (Batel et al., 2014; Jähnig et al., 2019; Rodin et al., 2023) including a separate nuclear Zeeman bath for all relevant nuclear species ($Z_i$), an electron non-Zeeman (eNZ) bath connected to a (virtual) cooling (CL) bath as well as the lattice, relaxing the different spin baths. During the build-up of hyperpolarization, the injection rate $k_{\mathrm{W}}$ describes the lumped contribution from the eNZ bath with its cooling and the subsequent transfer to the $Z_i$ bath. Upon switching off the microwave, the electron non-Zeeman bath remains only connected to the lattice and the nuclear spins. Specifically, considering Fig. 1 from Batel et al. (2014), switching off the microwave is equivalent to a vanishing cooling rate, leaving only the relaxation to the lattice for the $Z_i$ as well eNZ baths and the coupling between them. This leaves the system with only relaxation to the lattice remaining either directly from the $Z_i$ bath or through the eNZ bath. This joint relaxation process, as active during build-up and decay, is described through the relaxation rate $k_{\mathrm{R}}$ in the presented model.

It might be argued that the direct relaxation of nuclear spins to the lattice is a vanishing relaxation channel at dissolution DNP conditions as only paramagnetic relaxation is an effective relaxation mechanism under these conditions. For such a case, the bath model could be rewritten (and with a slight redefinition of the eNZ bath as composed of the electrons and the strongly hyperfine coupled nuclear spins) as very recently published (Rodin et al., 2023).

Cross effect (CE) DNP represents an intermediate effect which is described quantum mechanically but closely related with thermal mixing. Thus, it appears likely that the presented rate-equation model would be applicable for CE DNP too.

With this wide validity for DNP in mind, we are convinced that the model can be extended to other hyperpolarization methods. The interpretation of $k_{\mathrm{W}}$ and $k_{\mathrm{R}}$ for these scenarios is, however, beyond the scope of the current article.

## 7 Conclusions

We simulated and demonstrated experimentally the ability to correct for the effects of readout RF pulses in dynamic nuclear polarization. The proposed iterative correction approach allows to correct build-ups for up to 25° and decays for up to 12° RF flip angles. The experiments are supported by modelling based on a first-order differential equation which offers insights into the relationships between the experimental parameters of the balance between hyperpolarization injection and relaxation in experiments, eventually leading to a better understanding of the processes limiting the achievable hyperpolarization.

*Code and data availability.* All data and MATLAB scripts can be found under DOI:10.3929/ethz-b-000606640. A MATLAB script to perform the experimental flip angle corrections can be found under https://gitlab.ethz.ch/gvwitte/rfcorrection.

*Author contributions.* All authors designed the research. GvW developed the model and performed the simulations. GvW performed experiments and analysis. All authors discussed the results and were involved in writing the manuscript.

*Competing interests.* One of the (co-)authors (ME) is an executive editor and member of the editorial board of Magnetic Resonance. The peer-review process was guided by an independent editor, and the authors also have no other competing interests to declare.

*Acknowledgements.* We thank Aaron Himmler for help with the experiments and Gian-Marco Camenisch for help with sample preparation. We thank the anonymous referees for valuable feedback that improved the paper. ME acknowledges support by the Schweizerischer Nationalfonds zur Förderung der Wissenschaftlichen Forschung (grant no. 200020_188988). Financial support of the Horizon 2020 FETFLAG MetaboliQs grant is gratefully acknowledged.

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
