# Peer review of "Modelling and correcting the impact of RF pulses for continuous monitoring of hyperpolarized NMR"

_Magnetic Resonance, 2023_

## Referee Comment (RC2)

Modelling and correcting the impact of RF pulses for continuous monitoring of hyperpolarized NMR

Manuscript mr-2023-5

The manuscript summarizes work carried out to analyse the effect of radio frequency (rf) pulses on the polarization dynamics of samples in low temperature dynamics nuclear polarization experiments. The authors use a simple three parameter model to simulate a set of data assuming equally spaced rf pulses applied to monitoring the nuclear spin polarization buildup during microwave irradiation. Alternatively, they consider the decay of nuclear spin polarization after a non-thermal state was generated and the microwave irradiation was switch off. In their model they considered an electronic polarization reservoir, a rate by which electron spin polarization is converted into nuclear spin polarization and a rate by which nuclear spin polarization is lost by longitudinal relaxation processes. Furthermore, they added noise to the simulated data to generate data with a particular signal-to-noise ratio (SNR).

The simulated data were used to test three different correction scheme to correct for the polarization loss due to the application of the train rf pulses. The objective was to find differences in how well these correction schemes can recover the actual polarization buildup and decay rates, and the signal enhancement that are given by the choice of the simulation input parameters. The correction schemes used were i) a computational iterative scheme proposed by the authors (and based on their simple model) that provides corrected buildup time constant, longitudinal relaxation time constant and enhancement, an analytical correction formula published by Capozzi et al (2017) for the buildup time constant (in the manuscript called the CC method) and a correction based on the simple $\cos^n \theta$ formula, that corrects for the loss of nuclear spin polarization due to the application of rf-pulses with a pulse angle $\theta$.

Furthermore to these simulations, the three correction methods were applied to experimental data acquired with samples of 50mM 4-oxo-TEMPO in fully protonated water/glycerol or partly deuterated DNP juice.

Such an analysis is useful and should be interesting to a wider readership that uses samples with non-thermal polarization in their own research.

Assuming that I have correctly understood the data presented in tables 2 and 3 and Figs. 4 and 5, the key findings of the authors can be summarized by

(i)     All three correction schemes seem to work work well and give similar results for the longitudinal relaxation time constant provided that the pulse angle $\theta$ of the pulse train is kept reasonably small (e.g. 5 degree). For bigger excitation pulse angles, the simple $\cos^n \theta$ formula seem to fail when applied to experimental data for the DNP juice sample.

(ii)    The analytical formula of the CC method and the computational more demanding iterative scheme seem to give very similar results for the buildup time constants with, as claimed by the authors, the iterative scheme offering the additional advantage to also providing the corrected enhancement factor.

A number of questions arise mainly caused by the less-than-ideal presentation of the data in this manuscript:

i)      Fig. 2: It is not clear how much noise was added (see green dots) in these simulations (later in the discussion a SNR of 40 is quoted). The level of noise used should be added in the captions. Are all six subplots really necessary or is it not possible to provide the same information by just using 4 plots? The key information (apart from the obvious fact that data must be corrected to recover the true polarization dynamics) seems to be here that the iterative correction scheme tends to fails if noise is added to the model data and big rf pulse angles are used for the excitation. Therefore, it would be interesting to see how the quality of the iterative correction scheme depends on different added levels of noise and how this relates to the expected SNR in real experimental data. In the discussion, the authors conceded that they have assumed a much lower SNR (40) in their simulations in comparison to the SNR of 1000 obtained by a 2.4 degree excitation rf pulse for the fully protonated sample. Does this mean that the robustness of their iterative correction method

against low SNR is actually not really required since for experimental data the SNR is much higher than the assumed low value in the simulations?

ii)    Presentation of data in Fig. 3 should be improved: The captions do not explain the black curve in (a). Furthermore, in (b) it is very difficult to distinguish between the open diamond and open square box symbols. Why not using different colours instead ?

iii)   I suggest to move the lengthy tables 2 and 3 into an appendix or into the SI, since all the key information of these tables is already summarised in Fig. 4 and 5. In Fig. 5 (c) there seems to be excellent agreement for the relaxation time constant for all three correction methods when applied to the experimental data of the fully protonated sample (the existence of the green dots for the $\cos^n \theta$ correction scheme can only be inferred from the few green error bars in this subplot).

However, the $\cos^n \theta$ correction scheme seems to fail for smaller pulse angles when applied to the DNP juice sample (see Fig 4 c). In the discussion the authors imply that the SNR of the fully protonated sample is smaller than the SNR of the DNP juice sample due to the shorter FID caused by stronger dipolar broadening in the solid state sample. What is then the possible explanation that apparently the $\cos^n \theta$ correction scheme seems to fail when applied to correct for smaller pulse angles with a sample with higher SNR (i.e. the DNP juice sample) ?

It would be useful to include (e.g. in Tables 2 and 3) also the SNR (e.g. measured under steady state polarization conditions) of the two different sample depending on the pulse angles used.

Why is the symbol $\phi$ used for the pulse angle in the axes titles for Fig 4 and 5 instead of $\theta$ ?

Provided that the CC method gives a reliable correction for the buildup time constant, should it not be possible then to also obtain a reliable estimate of the enhancement value if the thermal polarization can be measured ?

iv)    The authors should consider using a more consistent terminology and rephrase some of their sentences. Instead of polarization 'injection', I suggest to use 'polarization transfer' and I would definitely avoid the use of 'RF relaxation-rate' and 'RF relaxation' (e.g. see page 10 ) since in NMR terminology relaxation is used to describe incoherent processes while the depletion or loss of nuclear spin polarization caused by application of rf pulses is usually an entirely coherent process.

v)     The information included in the SI document is very badly explained and is very difficult to understand by the interested reader without the addition of further details either in the captions of the supplementary figures or additional text in the SI document.

vi)    A more cosmetical detail: Please check the use of the multiplication symbol in your formulas. You have not consistently used it in all terms of your formulas ( e.g. see eqn 7 and 8). Its use is rather unconventional and I would delete it in all formulas.

In summary, while this manuscript covers some interesting aspects that need to be considered when analysing experimental data of non-thermal spin polarization dynamics, the current version requires significant improvements in both data presentation and careful analysis and discussion. In particular, a thorough analysis of how the various correction scheme depend on the SNR of the data would provide useful insight which scheme should be applied in which experimental scenario. In the current version no convincing arguments are made that the proposed iterative correction scheme for the buildup time constant would be superior to the already published CC method or that it would be superior to the $\cos^n \theta$ scheme for obtaining a properly corrected relaxation time constant.

/

---

## Author Response (AR2)

**Response to first review**

Dear Reviewer,

Thank you for your valuable feedback. Please find the answers to your comments (blue) below.

*In the introduction, a clear and easily comprehensible recap of the rate equations for the build-up dynamics is presented, and the basis for the iterative rf pulse correction algorithm is derived. In the beginning of the methods section the other pulse correction methods are presented (from my perspective these could have been already introduced in the preceding section), and the experimental procedure is explained in sufficient detail.*

We combined the description of all three correction methods in the RF correction section.

*The weakest point of the paper, however, is the presentation of the actual results. Particularly the experimental results (but to some degree also the simulation results) are only referred to as presented in the respective figures and tables, but no practically no explanation of the results is given. Here, I would have expected a more extensive presentation in the text, where the reader is guided through the rather large set of data in figs. and tables 2 and 3. From my perspective, a figure should support the presentation of the results in the text, and not be left alone to the reader to perform the interpretation on their own. In the discussion, the authors touch important points which could lead to the observed results or introduce errors, and in the conclusions a relatively short summary is given.*

We extended the previous description of the simulations in the main text to (page 7):

*"A comparison of simulated noise-free and noisy data without correction and with iterative correction is shown in Fig. 2. For larger flip angles, the uncorrected build-up deviates from the theoretical value without RF pulsing. Employing the iterative correction for noise-free data works accurately up to the largest flip angles tested (37°). Introducing noise into the simulations, leads to a small deviation for the largest flip angle considered. To study the performance of the corrections more systematically, we performed the corrections thousands of times for each $\theta$-TR pair considered. The results of these simulations are shown in the supplementary information (section S1 for build-ups, section S2 for decays), yielding similar accuracies and precisions for the CC-model and the iterative correction. The $1/\cos^{n-1}$ correction can perform similarly to the other two methods (see Discussion).*

*In addition to studying the accuracy and precision depending on the flip angle and repetition time, we simulated the SNR-dependence of the iterative and CC-model. For this, we varied the steady-state polarization as well as noise in 10 steps each and used all combinations thereof. The results for these are shown in Fig. 3. For both corrections a minimum SNR of around below 5 is found with slightly higher values for large flip angles (25°) to avoid build-up parameters deviating more than 10% from the values without RF pulsing. SNR in this context refers to the SNR at the steady-state of the uncorrected build-up."*

In the experimental results section, we added the following text (pages 9-11):

*"We first performed small flip angle measurements (for both samples separately) as for these the measured parameters are very close to the unperturbed case (cf. Tab. 1). After these calibration measurements, we performed measurements withlarger flip angles and different repetition times to estimate the range over which the corrections perform accurately*

*experimentally. The results for all measurements with TEMPO in DNP juice are summarized in Tab. 2 and Fig. 5. The respective data sets in the protonated sample are shown in Tab. 3 and Fig. 6, described by an apparent relaxation-rate due to RF perturbations, given by $\sin(\theta)/TR$.*

*For larger and more repeated pulses, the measured uncorrected parameters deviate strongly from calibration measurements. However, the corrected parameters give accurate values w.r.t. to the calibration measurements (±10%). In particular, build-ups can be corrected with the CC- and iterative correction up to 25° in our experiments. For the decay, the corrections become inaccurate earlier which will be discussed below: the $1/\cos^{(n-1)}$ becomes inaccurate for 5° pulses in our case (see Tab. 2), although its accuracy might be similar to the other two corrections with flip angles up to 12° possible (see Tab. 3). We note that the corrections can give accurate results with measured decay time constants of less than one-fifth of the expected value."*

Besides the above general criticism I have another more detailed question about the validity of the rate equation approach. The independence of buildup and decay rates which form the basis of Eq. (1) have been derived for solid effect DNP, where an additional polarization injection pathway is driven by microwave irradiation. Turning off the microwaves would then effectively set $k\_W$ to 0, leaving us only relaxation with $k\_R$ back to thermal equilibrium. Besides others, Shimon et al. (Phys. Chem. Chem. Phys., 2012, 14, 5729) have shown that under similar conditions, cross effect or thermal mixing may also be active. For these DNP mechanisms, polarization injection and relaxation cannot be separated as the nuclear polarization is always coupled to an electron polarization difference (or spectral gradient). Thus, the relaxation term should be small compared to $k\_W$ and turning off microwave irradiation would only modulate A to a value near thermal equilibrium, while $k\_W$ is unaffected. How would this affect the herein presented method? Can you make an estimate which DNP mechanism is dominant? From the observed difference in tau_bup and tau_rec, the assumption of having more or less independent injection and relaxation pathways seems obvious, but I am still wondering what cause some admixture of the other DNP mechanism (CE/TM) would have in the analysis.

The question regarding the validity of the model is very interesting. We believe that such a model can describe any type of DNP mechanism. We have updated the discussion to make clear that we do not assume any particular DNP mechanism in it. In addition, we have added a paragraph on pages 14/15 (discussion) which describes how the rate-equation model can be connected with the quantum mechanical description of the SE and CE as well as the thermodynamic description of TM:

*"For solid effect (SE) DNP the DNP injection into the bulk can be understood in terms of the polarization transfer from an electron to a hyperfine-coupled nucleus. From this polarized nucleus, the polarization spreads into the bulk through spin diffusion. The injection rate $k\_W$ can be understood as the overall rate for this joint process yielding a detectable magnetization creation in the bulk of the sample. Switching off the microwave would interrupt the hyperfine-mediated polarization transfer, causing a vanishing $k\_W$.*
 *Contrary to the quantum mechanical description of the solid effect, thermal mixing (TM) DNP is described in terms of a bath model with different temperatures for different spin systems. Such a spin bath model was used in previous work of our lab [Batel2014,Jähnig2019]: A separate nuclear Zeeman bath for all relevant nuclear species (Z_i), an electron non-Zeeman (eNZ) bath connected to a (virtual) cooling (CL) bath as well as the lattice, relaxing the baths. During the build-up of hyperpolarization, the injection rate $k\_W$ describes the lumped contribution from the eNZ bath with its cooling and the subsequent transfer to the Z_i bath. Upon switching off the microwave, the electron non-Zeeman bath remains only connected to the lattice and the nuclear*

*spins.*

*Specifically, considering Fig. 1 from Ref. [Batel2014], switching off the microwave is equivalent to a vanishing cooling rate, leaving only the relaxation to the lattice for the $Z_i$ as well eNZ baths and the coupling between them. This leaves the system with only relaxation to the lattice remaining either directly from the $Z_i$ bath or through the eNZ bath. This joint relaxation process, as active during build-up and decay, is described through the relaxation rate $k_R$ in the presented model.*

*It might be argued that the direct relaxation of nuclear spins to the lattice is a vanishing relaxation channel at dissolution DNP conditions as only paramagnetic relaxation is an effective relaxation mechanism under these conditions. For such a case, the bath model could be rewritten (and with a slight redefinition of the eNZ bath as composed of the electrons and the strongly hyperfine coupled nuclear spins) as very recently published in Ref. [Rodin2023].*

*Cross effect (CE) DNP represents an intermediate effect which is described quantum mechanically but closely related with thermal mixing. Thus, it appears likely that the presented rate equation model would be applicable for CE DNP too."*

*[Batel2014] Batel, … Ernst, PCCP 16 (2014)*

*[Jähnig2019] Jähnig, … Ernst; JMR 303 (2019)*

*[Rodin2023] Rodin, … Abergel; PCCP (2023)*

We believe that the discussion of the validity of the model and its connection to the more detailed understanding of the different DNP mechanisms adds significant value to the manuscript. We would like to thank the reviewer for this interesting question.

Minor remarks: the use of the multiplication sign is inconsistent, and oftentimes spaces after the degree symbol are either missing or repeated.

We removed the multiplication dots and checked the ° signs. Thank you for your careful reading.

**Response to second review**

Dear Reviewer,

Thank you for the critical evaluation of our manuscript and the detailed questions. We have tried to address them one-by-one. Your comments are inserted in blue.

i) Fig. 2: It is not clear how much noise was added (see green dots) in these simulations (later in the discussion a SNR of 40 is quoted). The level of noise used should be added in the captions. Are all six subplots really necessary or is it not possible to provide the same information by just using 4 plots?

We have reduced the number of subplots of the figure.

The key information (apart from the obvious fact that data must be corrected to recover the true polarization dynamics) seems to be here that the iterative correction scheme tends to fails if noise is added to the model data and big rf pulse angles are used for the excitation. Therefore, it would be interesting to see how the quality of the iterative correction scheme depends on different added levels of noise and how this relates to the expected SNR in real experimental data. In the discussion, the authors conceded that they have assumed a much lower SNR (40) in their simulations in comparison to the SNR of 1000 obtained by a 2.4 degree excitation rf pulse for the fully protonated sample. Does this mean that the robustness of their iterative correction method against low SNR is actually not really required since for experimental data the SNR is much higher than the assumed low value in the simulations?

We had a similar discussion about minimum SNR to perform the corrections after submitting the initial manuscript. We have added the results to the manuscript. We added a new figure for this in the manuscript together with a description thereof in the main text (pages 7/8):

*"In addition to studying the accuracy and precision depending on the flip angle and repetition time, we simulated the SNR-dependence of the iterative and CC-model. For this, we varied the steady-state polarization as well as noise in 10 steps each and used all combinations thereof. The results for these are shown in Fig. 3. For both corrections a minimum SNR of around below 5 is found with slightly higher values for large flip angles (25°) to avoid build-up parameters deviating more than 10% from the values without RF pulsing. SNR in this context refers to the SNR at the steady-state of the uncorrected build-up."*

ii) Presentation of data in Fig. 3 should be improved: The captions do not explain the black curve in (a). Furthermore, in (b) it is very difficult to distinguish between the open diamond and open square box symbols. Why not using different colours instead ?

The black curve is explained in the caption. We changed Fig. 2b.

iii) I suggest to move the lengthy tables 2 and 3 into an appendix or into the SI, since all the key information of these tables is already summarised in Fig. 4 and 5. In Fig. 5 (c) there seems to be excellent agreement for the relaxation time constant for all three correction methods when

applied to the experimental data of the fully protonated sample (the existence of the green dots for the $\cos^n \theta$ correction scheme can only be inferred from the few green error bars in this subplot).

The figures do not contain the full information as we used different repetition times for different flip angles, i.e. $\sin(\theta)/T_R$ is larger for 25° than for 37°. Additionally, for cases in which the corrections fail, not all the data is contained in the figures. For some readers it might be easier to understand the results based on the tables than on the figures. Thus, we have opted to keep the tables in the main part.

However, the $\cos^n \theta$ correction scheme seems to fail for smaller pulse angles when applied to the DNP juice sample (see Fig 4 c). In the discussion the authors imply that the SNR of the fully protonated sample is smaller than the SNR of the DNP juice sample due to the shorter FID caused by stronger dipolar broadening in the solid state sample. What is then the possible explanation that apparently the $\cos^n \theta$ correction scheme seems to fail when applied to correct for smaller pulse angles with a sample with higher SNR (i.e. the DNP juice sample) ?

We included a paragraph to address this into the discussion (page 13):

"The failure of the $1/\cos^{(n-1)}$ correction for the DNP juice sample already at 5° (compare Tab. 2) is partially related to the measurement process but mostly inherent to the single data point dependence of this correction. For DNP juice, the spin-lattice T1 relaxation time is much shorter than for the natural abundance sample. For both samples, we sampled until there was either only a thermal signal remaining or if several hundred seconds elapsed. If the signal approaches the thermal signal generated between subsequent acquisitions, the $1/\cos^{(n-1)}$ correction would give an increasing signal as the correction factor diverges while the signal remains constant. With a careful selection of the number of data points acquired or analyzed, this problem could be mitigated. The complete failure of the $1/\cos^{(n-1)}$ correction (Fig. S8) for flip angles of 25° and more can be explained as follows: If the decay is acquired much longer than the decay time under RF pulsing, many data points with only noise are acquired. This noise is subsequently amplified by the divergent correction factor, leading to signals much larger than at the beginning of the decay, spoiling the exponential fit of the data. Again, this could be mitigated with a careful selection of the number of data points. These problems are not encountered for the other corrections as these rely on a fit of the complete data set and do not require manual user selection of data points to be included into the analysis, representing a major advantage for the automatic analysis of larger data sets"

It would be useful to include (e.g. in Tables 2 and 3) also the SNR (e.g. measured under steady state polarization conditions) of the two different sample depending on the pulse angles used.

We considered this but opted not to include it as the tables are already fairly large and the informational benefit is relatively small. Examples for 2.4° acquisitions in the DNP juice as well as in the natural abundance sample are now contrasted in the SI.

Why is the symbol used for the pulse angle in the axes titles for Fig 4 and 5 instead of $\theta$ ?

We corrected this.

Provided that the CC method gives a reliable correction for the buildup time constant, should it not be possible then to also obtain a reliable estimate of the enhancement value if the thermal polarization can be measured ?

Originally, it was our idea to extend the CC-model, which treats the RF pulses as an additional rate reducing the hyperpolarization but we did not find a simple solution. The proposed iterative scheme could be implemented very easily based on the rate-equation model. After working on the above mentioned minimum SNR question, we found that the too low corrected parameters for the iterative correction (see Figs. 2, S1-S4) are not due to the noise but due to a problem with the self-consistent algorithm. Thus, we adjusted the iterative correction to perform the correction in a single step using the result of the CC-model as an input. This reduces the computational cost and the simulations give accurate results for all cases studied. With this in mind, it was possible to come up with an extension of the CC-model (page 6):

*"Based on our rate-equation model and the notion that the relative change of the steady-state polarization with RF pulsing is solely due to the change in build-up time (compare Eq. 4a and Tab. 1), the CC-model can be extended to provide corrected values for the steady-state polarization according to:*

$P_0 = P_0 \tau_{bup}/\tau$ ' (10)

*with τ' and P' being the measured, uncorrected build-up time and steady-state polarization and $\tau_{bup}$ the CC-corrected build-up time constant. Conceptually, this can be understood as the injection rate constant $k_W$ being undisturbed by the RF pulses while the relaxation-rate constant $k_R$ appears increased by RF pulsing."*

iv) The authors should consider using a more consistent terminology and rephrase some of their sentences. Instead of polarization 'injection', I suggest to use 'polarization transfer' and I would definitely avoid the use of 'RF relaxation-rate' and 'RF relaxation' (e.g. see page 10 ) since in NMR terminology relaxation is used to describe incoherent processes while the depletion or loss of nuclear spin polarization caused by application of rf pulses is usually an entirely coherent process.

We intentionally avoided the wording "polarization transfer" as this might often be mistaken as the hyperfine-mediated electron-nuclear spin transfer. However, this transfer might not relate well with the effective polarization build-up as e.g. spin diffusion alters the hyperpolarization build-up. Hence, we came up with the wording "polarization injection" to describe the polarization we add to the observable part of the sample.

We tried to be more careful in the updated version of the manuscript but we think it is important to understand the similarity between relaxation and RF pulses in hyperpolarization. In fact, this is the basic idea of the (now extended CC-correction, see above) as it treats the "RF-relaxation" as an additional relaxation channel. We tried to describe this carefully in the manuscript (page 6):

*"We note that relaxation in NMR usually describes incoherent processes, while RF pulses induce a coherent process. Assuming large hyperpolarization enhancements such that the thermal polarization can be neglected, incoherent spin lattice relaxation drives the polarization back to zero or more precisely to the (negligible) thermal equilibrium. Hence, RF pulses and incoherent relaxation processes have the same effect on the hyperpolarization. In the following, we will use the term "apparent relaxation due to RF perturbations" to refer to the polarization-depleting rate of RF pulses, indicating that they have a similar effect to spin-lattice relaxation in hyperpolarization but not being an incoherent relaxation process."*

We carefully used "apparent relaxation due to RF perturbations" throughout the manuscript.

v) The information included in the SI document is very badly explained and is very difficult to understand by the interested reader without the addition of further details either in the captions of the supplementary figures or additional text in the SI document.

We have updated the SI in terms of description as well as content. We agree that the current version of the SI might be too concise and hard to understand.

vi) A more cosmetical detail: Please check the use of the multiplication symbol in your formulas. You have not consistently used it in all terms of your formulas ( e.g. see eqn 7 and 8). Its use is rather unconventional and I would delete it in all formulas.

We removed all the multiplication symbols to avoid confusion.

In summary, while this manuscript covers some interesting aspects that need to be considered when analysing experimental data of non-thermal spin polarization dynamics, the current version requires significant improvements in both data presentation and careful analysis and discussion. In particular, a thorough analysis of how the various correction scheme depend on the SNR of the data would provide useful insight which scheme should be applied in which experimental scenario. In the current version no convincing arguments are made that the proposed iterative correction scheme for the buildup time constant would be superior to the already published CC method or that it would be superior to the cosn θ scheme for obtaining a properly corrected relaxation time constant.

It was not our intention to find a better correction for the time constants compared to the CC-model as this gives already accurate values. Instead, we wanted to find an approach to correct also the enhancements for the impact of RF pulses as the enhancement in many cases might be the more important quantity. Furthermore, since visual representation of data is essential in science, we intended to find an approach to show the corrected data, i.e. a build-up curve recorded with large flip angle, high SNR data but still visually showing the "true" build-up time and steady-state polarization. Additionally, we aimed at understanding over which range of flip angles such corrections could be used in experiments, justifying the use of the corrections even for flip angles between 10 and 20°, offering alternative approaches to measure low SNR materials. We hope that this, intentions become clearer in the revised manuscript.